# Google searches for suicide and suicide risk factors in the early stages of the COVID-19 pandemic

**Emily A. Halford**[1], **Alison M. Lake**[1], **Madelyn S. Gould**[1,2,3]*

**1** Division of Child and Adolescent Psychiatry, New York State Psychiatric Institute, New York, NY, United States of America, **2** Department of Epidemiology, Columbia University Mailman School of Public Health, New York, NY, United States of America, **3** Department of Psychiatry, Columbia University Vagelos College of Physicians and Surgeons, New York, NY, United States of America

* Madelyn.Gould@nyspi.columbia.edu

**Data Availability Statement:** All relevant data are within the manuscript and its Supporting Information files.

**Funding:** This project was supported by institutional funds provided by the Division of Child

## Abstract

A novel coronavirus (SARS-CoV-2), which causes the COVID-19 respiratory illness, emerged in December of 2019 and has since spread globally. The dramatic lifestyle changes and stressors associated with this pandemic pose a threat to mental health and have the potential to exacerbate risk factors for suicide. We used autoregressive integrated moving average (ARIMA) models to assess Google Trends data representing searches in the United States for 18 terms related to suicide and known suicide risk factors following the emergence of COVID-19. Although the relative proportion of Google searches for suicide-related queries was lower than predicted during the early pandemic period, searches for the following queries representative of financial difficulty were dramatically elevated: "I lost my job" (226%; 95%CI, 120%-333%), "laid off" (1164%; 95%CI, 395%-1932%), "unemployment" (1238%; 95%CI, 560%-1915%), and "furlough" (5717%; 95%CI, 2769%-8665%). Searches for the Disaster Distress Helpline, which was promoted as a source of help for those impacted by COVID-19, were also remarkably elevated (3021%; 95%CI, 873%-5169%). Google searches for other queries representative of help-seeking and general mental health concerns were moderately elevated. It appears that some indices of suicidality have fallen in the United States in this early stage of the pandemic, but that COVID-19 may have caused an increase in suicide risk factors that could yield long-term increases in suicidality and suicide rates.

## Introduction

A novel coronavirus (SARS-CoV-2), which causes the COVID-19 respiratory illness, emerged in Wuhan, China in December of 2019. On February 29, 2020, the first death attributed to COVID-19 within the United States occurred in Washington State.

As the virus has spread, the public health community has been rightfully focused on strained hospital systems and the rising death toll. However, COVID-19 poses a serious threat to mental health as well. Social distancing is one of the primary measures implemented to slow

and Adolescent Psychiatry, New York State Psychiatric Institute. There was no additional external funding received for this study.

**Competing interests:** The authors have declared that no competing interests exist.

the virus's spread, and while necessary to protect the public from COVID-19, this practice may have detrimental secondary effects on unemployment, loneliness, and pre-existing mental illnesses, all of which are known risk factors for suicide [1–5]. Additionally, the virus itself has the potential to exacerbate other suicide risk factors through a national sense of anxiety, grief as the virus claims more lives, and increased stress levels for essential workers [5, 6].

The aim of this analysis is to evaluate the impact of COVID-19 on suicidality in the United States during the pandemic's early stages, as well as the virus's effect on risk factors which might predict long-term impacts on suicide. Google Trends data was chosen for this study because it is far more rapidly available than alternative data sources such as mortality data, and because significant evidence exists that establishes an association between Google search behavior and actual suicidal behavior both within the United States [7–10] and internationally [11–16]. In fact, Google searches for "how to suicide," "how to kill yourself," and "painless suicide" were found to be better predictors of completed suicides in the United States than traditional self-reported measures of suicide risk [9]. Associations have also been found between searches for mental health diagnoses and other known risk factors for suicide and actual suicide rates [8, 11–13]. Searches for queries representing unemployment are frequently associated with suicide [8, 10, 11] and are of particular relevance to the present study given the far-reaching economic damage caused by the COVID-19 pandemic. Therefore, we assessed Google Trends data representing searches for terms related to suicide and known suicide risk factors following the emergence of COVID-19 in the United States with the aim of providing insight into how the early stages of this pandemic may have impacted suicidality and suicide rates.

## Methods

### Query selection

There were four primary domains which were examined due to their known associations with suicide risk [5]: suicide-specific queries, help-seeking queries, general mental health queries, and queries related to financial difficulty. We did not try to be exhaustive by including every possible search term that a user may have utilized within each category, but rather aimed to have 3 to 6 queries representative of each domain.

Queries were selected for the suicide-specific, general mental health, and financial difficulty categories based upon a review of the literature and use of the "Related Queries" tool available on the Google Trends interface. These finalized categories are as follows: suicide-specific queries ("suicide–squad" [11, 13, 14, 16], "kill myself," "kill yourself," "suicide methods + suicide method," "commit suicide" [7], and "fast suicide + easy suicide + quick suicide + painless suicide" [9]), general mental health queries ("depression" [11], "panic attack," and "anxiety" [8, 11–13]), queries related to financial difficulty ("unemployment" [10, 11], "I lost my job," "laid off" [8], "furlough" (a mandatory temporary leave for employees), and "loan"). The uncategorized query "loneliness" was also included to assess possible impacts of the social distancing interventions implemented to slow the spread of COVID-19.

Queries for the help-seeking category were intended to represent the two most prominent and widely-used crisis services in the United States (National Suicide Prevention Lifeline, the national network of local crisis centers, https://suicidepreventionlifeline.org/; Crisis Text Line, the global organization providing crisis intervention over text message, https://www.crisistextline.org/), as well as the Disaster Distress Helpline, a national hotline providing crisis counseling for those impacted by any natural or human-caused disaster that has been promoted during the COVID-19 pandemic (https://www.samhsa.gov/find-help/disaster-distress-helpline). This finalized category contains the following queries: "crisis text line," "national

suicide prevention lifeline + national suicide hotline," and "disaster distress helpline + disaster distress hotline."

Google Trends (https://trends.google.com/trends/?geo=US) allows queries containing certain words to be excluded from the relative search proportion using the "-"sign, and searches for the popular movie *Suicide Squad* were thus excluded from our "suicide" search query by requesting data for "suicide–squad" [17]. We considered using this feature to filter out searches for "depression" that included "economy," "economic," and "are we going into a" to help ensure that we captured searches related to emotional depression rather than an economic depression. However, doing so made no meaningful change in observed search volume so we decided to analyze the simple "depression" query used in three earlier studies [11, 13, 14]. Google trends also allows for the relative search volume associated with several search queries to be combined using the "+" sign. This tool was used to yield a combined relative search proportion for common variations on terms or conceptually linked terms, resulting in the following queries: "national suicide prevention lifeline + national suicide hotline," "disaster distress helpline + disaster distress hotline," "suicide methods + suicide method," and "fast suicide + easy suicide + quick suicide + painless suicide."

## Data

The gtrendsR package in R was used to retrieve weekly search data from March 3, 2019 through April 18, 2020 for 14 of the 18 search queries. The current version of the gtrendsR package lacks the additive "+" feature, so data associated with the following 4 queries were obtained directly from the Google Trends website (https://trends.google.com/trends/?geo=US): "national suicide prevention lifeline + national suicide hotline," "disaster distress helpline + disaster distress hotline," "suicide methods + suicide method," and "fast suicide + easy suicide + quick suicide + painless suicide."

Data for each query is not an absolute number of searches or an absolute proportion; rather data for each query is rescaled to represent a relative search proportion. Weekly search volume for a given query is first divided by total Google search volume for that week. The proportions are then scaled from 0 to 100, with 100 assigned to the week within the designated data collection period (in this case, March 3, 2019-April 18, 2020) and specified location (in this case, the United States) when the given query's proportion of the total search volume was greatest. A relative proportion of 25, for example, indicates a proportion 25% as high as the given query's highest relative proportion in the study period.

Relative search volume for a given query includes all Google searches containing the specific queried words, with or without additional words. For example, searches for "how to kill myself" would be included in the search volume for our query "kill myself."

## Time period and location

Data was limited to Google searches conducted within the United States between March 3, 2019 and April 18, 2020. The first death attributed to COVID-19 within the United States occurred on February 29, 2020, and the following day (March 1, 2020) was selected as the beginning of the early pandemic period to reflect the shift in public perception that was associated with this event. One year of data (March 3, 2019 –February 29, 2020) was used to train the ARIMA models as longer time periods would include surges in suicide-related queries associated with unrelated events such as the deaths by suicide of Kate Spade and Anthony Bourdain. Given that Google Trends provides weekly data for time intervals of this length, March 3, 2019 represents the beginning of the week as close as possible to falling exactly one year prior to the beginning of the early pandemic period beginning March 1, 2020.

### Analytic strategy

Autoregressive integrated moving average (ARIMA) modeling was used to obtain predicted weekly relative search volume for each query for the March 1, 2020 to April 18, 2020 early pandemic period (Supplement 1). The four aforementioned query categories were entirely conceptual and queries were not combined during the model-building stage. Queries were therefore analyzed separately, yielding 18 total models constructed using search volume relative to all Google searches within the United States within the study period. Weekly search data from the preceding year (March 3, 2019 through February 29, 2020) was used to train the ARIMA models. Percent difference was calculated between this predicted relative search proportion and observed relative search proportion for each query at each week of the study period. Mean percent difference for each query with bootstrap 95% confidence intervals were then calculated for the full study period. All analyses were conducted in R version 3.6.1.

## Results

Average relative search volume associated with four of the six suicide-specific search queries was lower than predicted during the study period (Fig 1). These queries were "suicide method + suicide methods" (-32%; 95%CI, -40% to -24%), "suicide–squad" (-20%; 95%CI, -27% to -14%), "commit suicide" (-18%; 95%CI, -24% to -12%), and "kill yourself" (-10%; 95%CI, -16% to -5%). Confidence intervals for mean percent difference associated with the remaining 2 suicide-related queries ("fast suicide + easy suicide + quick suicide" + painless suicide and "kill myself") contained zero.

Among the queries indicative of general mental health, average relative search volume was elevated for "depression" (3%; 95%CI, 1%-6%) and "panic attack" (12%; 95%CI, 2%-21%). Relative search volume for "anxiety" did not differ from predicted volume.

Average relative search volume for "loneliness" was 24% greater than expected (95%CI, 12%-37%).

Average relative search volume was elevated for all queries indicative of help-seeking, including "national suicide prevention lifeline + national suicide hotline" (9%; 95%CI, 1%-18%), "crisis text line" (36%; 95%CI, 18%-54%), and "disaster distress helpline + disaster distress hotline" which experienced a large surge in search volume with average relative search volume reaching 3021% greater than expected (95%CI, 873%-5169%).

Average relative search volume was also sharply elevated for every query indicative of financial difficulty, including "loan" (37%; 95%CI, 17%-58%), "I lost my job" (226%; 95%CI, 120%-333%), "laid off" (1164%; 95%CI, 395%-1932%), "unemployment" (1238%; 95%CI, 560%-1915%), and "furlough" (5717%; 95%CI, 2769%-8665%).

## Discussion

Google search behavior appears to have dramatically changed following the emergence of COVID-19 in the United States, particularly for queries related to financial difficulty ("I lost my job," 226%; "laid off," 1164%; "unemployment," 1238%; "furlough," 5717%) and the Disaster Distress Helpline ("disaster distress helpline + disaster distress hotline," 3021%). Searches related to suicide are lower than expected, while searches related to general mental health and help-seeking are moderately elevated. Search volumes for suicide-specific terms such as "suicide," "commit suicide," and "kill yourself," which have previously been shown to have a positive correlation with suicide rates [7, 9, 11] are all lower than predicted. However, searches for queries representative of financial difficulty have also been shown to have a positive association with suicide rates [8, 10, 11] and the scale at which search volume for these queries increased in the present study is remarkable. This marked increase in search volume is particularly concerning considering that out of eighty-nine search queries evaluated for correlation with

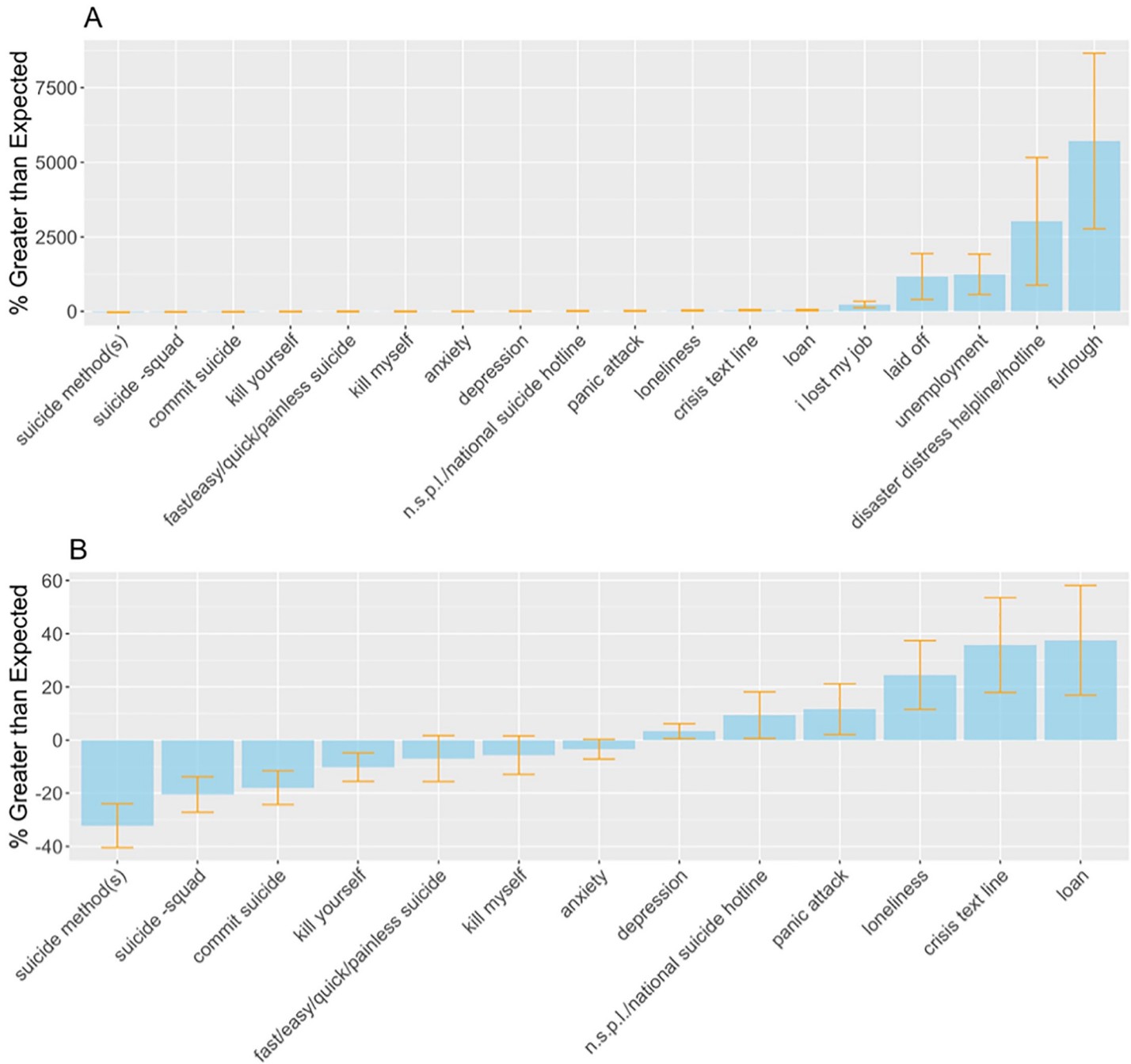

**Fig 1. Google searches between March 1 and April 18, 2020.** (A) Average percent difference between observed relative search proportion and relative search proportion predicted with pre-COVID-19 data. Percentage greater than expected shown for all search queries with 95% confidence intervals. (B) Data presented is the same as in panel A, but limited to search queries with percent greater than expected. <|100%|.

monthly suicide rates in the United States between 2004 and 2017, the strongest positive correlation was found for the query "laid off" with a lag time of 2 months [8]. It seems as though individuals are grappling with the immediate stresses of job loss and isolation and are consequently reaching out for help from crisis services, but that the impact on suicidality has not yet occurred.

There is significant cause for concern regarding the long-term impact of COVID-19 on suicide in the United States. While it is common for suicide rates to decrease in the immediate aftermath of national crises such as the September 11 attacks [18], there is evidence of long-term increases in suicide rates due to the 1918 flu pandemic [19] and the 2003 SARS outbreak in Hong-Kong [20]. Additionally, COVID-19 and the national response to the virus affect many suicide risk factors in ways that have been described as a "perfect storm" for suicide [5]. Increased isolation, unhealthy home-life situations, financial difficulties, grief associated with the death of loved ones, and other exacerbated risk factors may lead to an increase in suicidality and suicide attempts in the future. Many of these suicide predictors have already seen meaningful changes reflected in Google Trends data, such as an apparent increase in searches potentially representative of unemployment, panic attacks, loneliness, and the need for crisis hotlines.

Although we cannot know for certain the motivation or context behind any particular Google search, alternative data sources corroborate the patterns shown in the search data. Unemployment claims in the United States were over 20 times higher during the week ending on April 18, 2020 than during the comparable week in 2019 (4,267,395 compared to 211,762) and over 26 million claims have been filed since the start of the pandemic [21]. Additionally, although weekly calls to the National Suicide Prevention Lifeline have remained stable, calls to the Disaster Distress Helpline during the early pandemic period increased 1073% relative to the equivalent weeks in 2019 (personal communication, Sean Murphy PhD., Vibrant Emotional Health, administrative agency for these crisis services). As such, the increase in Google searches for the Disaster Distress Helpline is correlated with an increase in actual call volume, a potential indicator of the success of the promotion efforts for this service during the COVID-19 pandemic.

Although we do not have data regarding absolute changes in demand for Crisis Text Line services since the start of the pandemic, data insights published on the organization's website (www.crisistextline.org) are closely aligned with the trends observed in Google search behavior. The Crisis Text Line reports increases in conversations about grief, the coronavirus disease, loneliness, sexual assault and emotional abuse during quarantine, anxiety, depression, financial concerns, and stress associated with frontline work. Interestingly, reports of suicidal ideation have simultaneously fallen from 28% of chats to 22% of chats. Increases in Google searches for "crisis text line" indicate that more people may be reaching out to this service for support, and insights from these Crisis Text Line conversations support the hypothesis that Americans are seeking help for the serious emotional difficulties associated with COVID-19 while experiencing less suicidality.

The present study has several important limitations. Google search data is particularly prone to selection bias as access to and inclination to use search engines likely differ substantially between sub-populations. Additionally, lack of data on actual suicidal behavior prohibits firm conclusions from being drawn regarding the predictive capabilities of the search queries included in this study. Inclusion of mortality data would also allow for sub-group analyses, which are crucially important given the differing relationships between search behavior and suicidal behavior previously observed based upon age and gender [8, 9, 11, 13, 15]. Once 2020 mortality data becomes available, the present study can be built upon by assessing the relationship between the changes in search behavior observed in the present study and changes in suicide rates within the United States. Finally, we cannot be sure how the Google search results for any of the queries assessed in this study may mediate a relationship between search behavior and suicidality or suicide attempts. For example, the telephone number for the National Suicide Prevention Lifeline is the first result displayed for many queries associated with suicidal ideation, such as "kill myself," and search results highlighting such sources of help may impact post-search behavior.

The results of the current study highlight the ongoing emotional and financial struggles of Americans in the face of the COVID-19 pandemic. Although relative search volume for explicitly suicide-related queries has dropped in these early stages of the pandemic's progression, long-term suicidality is an important concern given the impact of COVID-19 on known risk factors for suicide.

## Supporting information

**S1 File. Weekly observed and predicted relative Google search proportion.**
(DOCX)

## Acknowledgments

We would like to acknowledge Dr. Hanga Galfalvy, Associate Professor of Biostatistics (in Psychiatry) at the Columbia University Medical Center, for her consultation on the analyses.

## Author Contributions

**Conceptualization:** Emily A. Halford, Alison M. Lake, Madelyn S. Gould.

**Data curation:** Emily A. Halford, Alison M. Lake, Madelyn S. Gould.

**Formal analysis:** Emily A. Halford.

**Funding acquisition:** Madelyn S. Gould.

**Investigation:** Emily A. Halford, Alison M. Lake, Madelyn S. Gould.

**Methodology:** Emily A. Halford, Alison M. Lake, Madelyn S. Gould.

**Project administration:** Emily A. Halford, Alison M. Lake, Madelyn S. Gould.

**Resources:** Madelyn S. Gould.

**Software:** Emily A. Halford, Alison M. Lake, Madelyn S. Gould.

**Supervision:** Alison M. Lake, Madelyn S. Gould.

**Validation:** Emily A. Halford, Alison M. Lake, Madelyn S. Gould.

**Visualization:** Emily A. Halford, Alison M. Lake.

**Writing – original draft:** Emily A. Halford.

**Writing – review & editing:** Alison M. Lake, Madelyn S. Gould.

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
