## [Decision Letter · Decision Letter 0]

23 Jun 2020

PONE-D-20-12386

Google searches for suicide and suicide risk factors in the early stages of the COVID-19 pandemic

PLOS ONE

Dear Dr. Gould,

Thank you for submitting your manuscript to PLOS ONE. After careful consideration, we feel that it has merit but does not fully meet PLOS ONE’s publication criteria as it currently stands. Therefore, we invite you to submit a revised version of the manuscript that addresses the points raised during the review process.

As the reviewers note below, some elements of the methodology are unclear or require revision, and in some cases the reviewers have suggested clear ways to address the concerns. Additionally, the paper would benefit from more careful interpretation, in particular regarding the extent to which internet searches are indicative of offline behavior.

We look forward to receiving your revised manuscript.

Kind regards,

Neal Doran

Academic Editor

PLOS ONE

Journal Requirements:

"This project was partially supported from institutional funds provided by the Division of Child and Adolescent Psychiatry, New York State Psychiatric Institute"

Reviewers' comments:

Reviewer's Responses to Questions

**Comments to the Author**

1. Is the manuscript technically sound, and do the data support the conclusions?

Reviewer #1: Partly

Reviewer #2: Partly

2. Has the statistical analysis been performed appropriately and rigorously? 

Reviewer #1: Yes

Reviewer #2: Yes

3. Have the authors made all data underlying the findings in their manuscript fully available?

Reviewer #1: Yes

Reviewer #2: Yes

4. Is the manuscript presented in an intelligible fashion and written in standard English?

Reviewer #1: Yes

Reviewer #2: Yes

5. Review Comments to the Author

Reviewer #1: In the introduction the authors could also refer to previous work that has shown that Google search behavior is in fact related to future suicidality (e.g., Arendt & Scherr, 2017a, 2017b).

Search terms such as “depression” can be regarded ambiguous given the economic impact of COVID-19 and discussions about whether the pandemic drove into an economic depression. It would be advisable to rule out such impact by also including (i.e., controlling for) economy related search terms at the same time in models in order to rule out the ambiguity of certain search terms. Interestingly, this has been done with the term “suicide -squad” that I assume in many cases refers to the movie and has therefore been explicitly excluded in order to not make the other query shares “disappear” relative to it. This would also be a strategy to handle other ambiguity around depression related search terms.

What was the rationale for looking at the specific time interval from March 3 to April 18? Please clarify.

In the results section, for some of the search terms the percentage sign is missing (p. 6, l. 146-147). Are the Google trends numbers calculated within each group of queries or have they all been lumped together and are presented relative to all? This information needs to be updated and included in the methods section.

The discussion section should refrain from giving the impression that the Google searches can automatically be translated into actual behaviors. More nuance would be good. Moreover, for certain search terms, certain languages, and only in certain countries Google suggests suicide prevention resources to some users in form of a Suicide Prevention Result (SPR) (see Haim et al., 2019; Scherr et al., 2019; Arendt et al., 2020), which is likely to yield differential (protective) effects on post-search behaviors (i.e., not the same effects for all across the board). This could be discussed in order to bring more nuance to the findings, or to explain counterintuitive findings. It might actually be that google increased the display frequency in some countries as a reaction to COVID-19.

Although Google trends data can be a good indicator for actual behavior especially in the suicide domain (Arendt & Scherr, 2017a, 2017b), conclusions cannot be drawn between the search queries given that we don't even know in how far there is an overlap of people who googled for these different groups of search queries (i.e., problem of an ecological fallacy). Above and beyond that, it is also worthwhile to discuss the opposite effect direction in that people with a higher vulnerability are stressed more (see Scherr, Toma, & Schuster, 2019) through the corona crisis, and therefore Google for mental health resources or suicide-related terms in order to find support (see e.g., Scherr & Reinemann, 2016) not the other way round.

Reviewer #2: Thank you for the opportunity to review this paper. It is a clear, well-written and interesting piece exploring trends in Google searches for suicide-related terms during the COVID-19 pandemic, and the results suggest some interesting trends in behaviour during this time. While overall it appears to be a sound piece of work, I have several suggestions that may improve its clarity.

1) Firstly, it would be useful to explain in the introduction why you chose to look at Google searches as your indicator of suicidality. Why was this chosen above other indicators such as, for example, the number of calls to mental health helplines, number of admissions to emergency departments for suicide attempts, etc.? How reliable an indicator is the number of Google searches e.g. what is the evidence that an increase in Google searches corresponds to an increase in behaviour? Providing this context in the introduction will strengthen your paper and the implications of your findings.

2) How did you decide on your search terms? A short description of this process would be useful to justify your chosen terms. It is likely that people experiencing suicidality may not include the word “suicide” in their Google search, and may instead use more general terms (e.g. “painless death” rather than “painless suicide”).

3) In the results section, it would be useful to discuss the magnitude of your effects (some of which are considerable). At present, this section reports the search terms that had an increase during the period of the study, but not the size of this increase. The variation here is quite large and worthy of comment – it ranges from a 3% increase for “depression” through to over 5700% for “furlough,” yet both of these terms are discussed as if the findings were relatively equal. I would suggest that the magnitude of the effect for some of these terms is a bigger finding than merely the fact that there was an increase. Similarly, the increase in searches for “Disaster Distress Helpline” and “Crisis Text Line” are mentioned in the discussion section as though they are relatively equal, although one saw an increase of 36% and the other and increase of over 3000%.

4) It would be useful to explain some of your terms for an international audience – although I understand that your study was focused on America, I was not familiar with some of the terms and services described. In particular, the Disaster Distress Helpline and Crisis Text Line receive some attention in the discussion section, but I was not sure what these services are or how they differ from each other. I also believe that “furlough” may be an American term, and considering its effect size it would be useful to make sure that international readers understand what it means.

5) Similarly to my first comment, your discussion section would be greatly strengthened if a little more evidence for your methods was provided. For example, is there evidence to support your claim that a drop in suicide-related terms equates to a drop in suicidality during the study period? This is implied, but if it could be supported by evidence it would make your claims stronger. If not, adding this as a limitation would be worthwhile.

6. PLOS authors have the option to publish the peer review history of their article (what does this mean?). If published, this will include your full peer review and any attached files.

Reviewer #1: No

Reviewer #2: No

---

## [Author Response · Author response to Decision Letter 0]

1 Jul 2020

Dear Dr. Doran:

We greatly appreciate the thorough review of our manuscript. We have revised the manuscript

to address the journal requirements and each concern raised by the two reviewers. Our

responses to each of the suggestions are described below. For your convenience, each of the

reviewers’ suggestions precedes our response, and we have provided page and line numbers to 

facilitate your finding the revisions in the manuscript. These page and line numbers correspond

to those in the version of the manuscript with changes highlighted using the track changes

mode in MS Word.

Journal Requirements:

1. Please ensure that your manuscript meets PLOS ONE’s style requirements, including those for file naming.

Files have been renamed to meet PLOS ONE’s style requirements (see Fig1.tiff and S1_File.pdf). 

2. Please include captions for your Supporting Information files at the end of your manuscript, and update any in-text citations to match accordingly. 

Captions have been added for the Supporting Information file at the end of the manuscript (see p. 15). 

Our amended funding statement is as follows: This project was supported by institutional funds provided by the Division of Child and Adolescent Psychiatry, New York State Psychiatric Institute. There was no additional external funding received for this study. 

Concerns of Reviewer 1:

1. In the introduction the authors could also refer to previous work that has shown that Google search behavior is in fact related to future suicidality.

Reviewer 2 has expressed a similar concern. We have added references to previous work evaluating the relationship between Google search behavior and observed suicide rates to the Introduction and Discussion sections. 

Specifically, we have added the following passage to the Introduction (p. 3, l. 87 – p. 4, l. 99): 

“Google Trends data was chosen for this study because it is far more rapidly available than alternative data sources such as mortality data, and because significant evidence exists that establishes an association between Google search behavior and actual suicidal behavior both within the United States (Gunn & Lester 2013; Lee 2020; Ma-Kellams et al. 2016; Parker et al. 2017) and internationally (Arora et al. 2016; Yang et al. 2011; Barros et al. 2019; Kristoufek et al. 2016; Hagihara & Miyazaki 2012; Song et al. 2014). In fact, Ma-Kellams et al. (2016) found that Google searches for “how to suicide,” “how to kill yourself,” and “painless suicide” were better predictors of completed suicides in the United States than traditional self-reported measures of suicide risk. Associations have also been found between searches for mental health diagnoses and other known risk factors for suicide and actual suicide rates (Arora et al. 2016; Yang et al. 2011; Barros et al. 2019; Lee et al. 2020). Searches for queries representing unemployment are frequently associated with suicide (Arora et al. 2016; Lee 2020; Parker et al. 2017)…..“

We have added the following passage to the Discussion (p. 10, l. 250 – p. 11, l. 260):

“Search volumes for suicide-specific terms such as “suicide,” “commit suicide,” and “kill yourself,” which have previously been shown to have a positive correlation with suicide rates (Arora et al. 2016; Gunn & Lester 2013; Ma-Kellams et al. 2016) are all lower than predicted. However, searches for queries representative of financial difficulty have also been shown to have a positive association with suicide rates (Arora et al. 2016; Parker et al. 2017; Lee 2020) and the scale at which search volume for these queries increased in the present study is remarkable. This marked increase in search volume is particularly concerning considering that out of eighty-nine search queries evaluated for correlation with monthly suicide rates in the United States between 2004 and 2017, the strongest positive correlation was found for the query “laid off” with a lag time of 2 months (Lee 2020).”

2. Search terms such as “depression” can be regarded ambiguous given the economic impact of COVID-19 and discussions about whether the pandemic drove into an economic depression. It would be advisable to rule out such impact by also including (i.e., controlling for) economy related search terms at the same time in models in order to rule out the ambiguity of certain search terms. Interestingly, this has been done with the term “suicide –squad” that I assume in many cases refers to the movie and has therefore been explicitly excluded in order to not make the other query shares “disappear” relative to it. This would also be a strategy to handle other ambiguity around depression related search terms. 

In light of this question, we conducted three additional analyses subtracting out the words “economy,” “economic,” and “are we going into a” from the “depression” query. Overall search volume and the patterns in search volume during the study period both remained almost identical when removing any of these three phrases from the “depression” query, indicating that very few searches which clearly indicated an interest in an economic depression were contributing to our results. Additionally, the change in search volume for the “depression” query was quite small in our study, and if it were truly confounded with economic depression we would likely see a much greater percentage change given the enormous changes observed for the other economic terms such as “unemployment.” We therefore decided that it was justifiable to leave the “depression” query as is, and to not specifically select out these economic terms. This information has been added to the Methods section (see p. 6, l. 140 – l. 145), in a new section (Query selection) that more thoroughly describes our query selection process. In regards to the depression query, we specified: 

“We considered using this feature to filter out searches for “depression” that included “economy,” “economic,” and “are we going into a” to help ensure that we captured searches related to emotional depression rather than an economic depression. However, doing so made no meaningful change in observed search volume so we decided to analyze the simple “depression” query used by Arora et al. (2016), Barros et al. (2019), and Kristoufek et al. (2016).” 

3. What was the rationale for looking at the specific time interval from March 3 to April 18? Please clarify.

We first want to clarify that this study was limited to searches within the United States. The first death attributed to COVID-19 in the United States occurred on February 29, 2020, and we selected the following day (March 1, 2020) as the beginning of our analysis period given the shift in public perception of the virus that we witnessed occurring around that time. April 18 was chosen as the end of the study period because search data for later dates was not available to us when we conducted our analyses in late April. We decided to use one year of training data to train the ARIMA models as two years of data would include significant peaks in suicide-related queries associated with the deaths by suicide of Kate Spade and Anthony Bourdain. Given the weekly format of Google Trends data for periods longer than 90 days, March 3 represents the beginning of the week that was as close as possible to being exactly one year prior to March 1st. 

The methods section has been clarified and an additional “Time period and location” section (see p. 8, l. 189 – l. 199) has been added to elaborate on the selection of these dates. 

4. In the results section, for some of the search terms the percentage sign is missing (p. 6, l. 146-147).

Percentage signs have been added where appropriate on these lines in the Results section (now p. 10, l. 241) in the revised manuscript). 

5. Are the Google trends numbers calculated within each group of queries or have they all been lumped together and are presented relative to all? This information needs to be updated and included in the methods section. 

Each query was analyzed separately, so all are presented relative to all Google searches made in the United States during our study period. This has been clarified in the Methods section (see p. 8, l. 204 – p. 9, l. 207) with the following statement: 

“The four aforementioned query categories were entirely conceptual and queries were not combined during the model-building stage. Queries were therefore analyzed separately, yielding 18 total models constructed using search volume relative to all Google searches within the United States within the study period. ”

6. The discussion section should refrain from giving the impression that the Google searches can automatically be translated into actual behaviors. More nuance would be good. Moreover, for certain search terms, certain languages, and only in certain countries Google suggests suicide prevention resources to some users in form of a Suicide Prevention Result (SPR) (see Haim et al., 2019; Scherr et al., 2019; Arendt et al., 2020), which is likely to yield differential (protective) effects on post-search behaviors (i.e., not the same effects for all across the board). This could be discussed in order to bring more nuance to the findings, or to explain counterintuitive findings. It might actually be that google increased the display frequency in some countries as a reaction to COVID-19. Although Google trends data can be a good indicator for actual behavior especially in the suicide domain (Arendt & Scherr, 2017a, 2017b), conclusions cannot be drawn between the search queries given that we don't even know in how far there is an overlap of people who googled for these different groups of search queries (i.e., problem of an ecological fallacy). Above and beyond that, it is also worthwhile to discuss the opposite effect direction in that people with a higher vulnerability are stressed more (see Scherr, Toma, & Schuster, 2019) through the corona crisis, and therefore Google for mental health resources or suicide-related terms in order to find support (see e.g., Scherr & Reinemann, 2016) not the other way round.

We have added depth and nuance to the discussion section. The existing evidence supporting a relationship between Google search behavior and suicide deaths has been readdressed in the discussion, and we have clarified that ambiguity remains regarding the true association between Google search behavior and suicidality during the COVID-19 pandemic. 

We have added a limitations section to the discussion (p. 13, l. 303 – l. 318) which describes our inability to draw conclusions from Google Trends data, as well as the possible mediating effect of Google’s Suicide Prevention Result on the relationship between searches and suicidal behavior. See below:

“The present study has several important limitations. Google search data is particularly prone to selection bias as access to and inclination to use search engines likely differ substantially between sub-populations. Additionally, lack of data on actual suicidal behavior prohibits firm conclusions from being drawn regarding the predictive capabilities of the search queries included in this study. Inclusion of mortality data would also allow for sub-group analyses, which are crucially important given the differing relationships between search behavior and suicidal behavior previously observed based upon age and gender (Arora et al. 2016; Barros et al. 2019; Lee 2020; Ma-Kellams et al. 2016; Hagihara & Miyazaki 2012). Once 2020 mortality data becomes available, the present study can be built upon by assessing the relationship between the changes in search behavior observed in the present study and changes in suicide rates within the United States. Finally, we cannot be sure how the Google search results for any of the queries assessed in this study may mediate a relationship between search behavior and suicidality or suicide attempts. For example, the telephone number for the National Suicide Prevention Lifeline is the first result displayed for many queries associated with suicidal ideation, such as “kill myself,” and search results highlighting such sources of help may impact post-search behavior.”

Concerns of Reviewer 2:

1. Firstly, it would be useful to explain in the introduction why you chose to look at Google searches as your indicator of suicidality. Why was this chosen above other indicators such as, for example, the number of calls to mental health helplines, number of admissions to emergency departments for suicide attempts, etc.? How reliable an indicator is the number of Google searches e.g. what is the evidence that an increase in Google searches corresponds to an increase in behaviour? Providing this context in the introduction will strengthen your paper and the implications of your findings.

Google search data was chosen as an indicator of suicidality because it is far more rapidly available than most other data sources. Additionally, many emergency departments were so overloaded with COVID-19 patients that non-COVID-19 patients, including suicidal individuals, had difficulty accessing care or postponed treatment in expectation of issues accessing care. It would therefore have been difficult to accurately assess suicidality using that measure, as suicidality is probably very underrepresented by ER data right now. While calls to helplines certainly represent an important data source (and changes in call volume to two such helplines are referenced in the Discussion section), they are primarily an indicator of help-seeking behavior and we were also interested in suicide risk and economic indicators in particular.

As noted in our reply to the first concern raised by Reviewer 1, significant evidence exists that establishes a relationship between Google search behavior and national suicide rates in several countries. This evidence, as well as our general rationale for using Google searches, has been expanded upon in our Introduction section (see p. 3, l. 87 – p. 4, l. 99) and is also provided below for your convenience: 

“Google Trends data was chosen for this study because it is far more rapidly available than alternative data sources such as mortality data, and because significant evidence exists that establishes an association between Google search behavior and actual suicidal behavior both within the United States (Gunn & Lester 2013; Lee 2020; Ma-Kellams et al. 2016; Parker et al. 2017) and internationally (Arora et al. 2016; Yang et al. 2011; Barros et al. 2019; Kristoufek et al. 2016; Hagihara & Miyazaki 2012; Song et al. 2014). In fact, Ma-Kellams et al. (2016) found that Google searches for “how to suicide,” “how to kill yourself,” and “painless suicide” were better predictors of completed suicides in the United States than traditional self-reported measures of suicide risk. Associations have also been found between searches for mental health diagnoses and other known risk factors for suicide and actual suicide rates (Arora et al. 2016; Yang et al. 2011; Barros et al. 2019; Lee et al. 2020). Searches for queries representing unemployment are frequently associated with suicide (Arora et al. 2016; Lee 2020; Parker et al. 2017), and are of particular relevance to the present study given the far-reaching economic damage caused by the COVID-19 pandemic. Therefore, we assessed Google Trends data representing searches for terms related to suicide and known suicide risk factors following the emergence of COVID-19 in the United States with the aim of providing insight into how the early stages of this pandemic may have impacted suicidality and suicide rates.” 

We also expanded our Discussion to highlight the association between Google searches and suicidal behavior. Please see our response to your fifth concern for details. 

2. How did you decide on your search terms? A short description of this process would be useful to justify your chosen terms. It is likely that people experiencing suicidality may not include the word “suicide” in their Google search, and may instead use more general terms (e.g. “painless death” rather than “painless suicide”).

The categories of explicitly suicide-related queries, help-seeking queries, queries related to financial difficulty, and general mental health problems were selected before the individual queries themselves were decided upon. These categories were chosen due to their known association with suicide risk (Reger et al. 2020). 

Queries for the explicitly suicide-related, financial difficulty, and general mental health problems were selected through a literature review and use of the “Related queries” tool on the Google trends website. Queries for the help-seeking category were designed to represent the National Suicide Prevention Lifeline, the Crisis Text Line, and the Disaster Distress Helpline. 

A separate section (“Query selection,” see p. 4, l. 109 – p. 6, l. 150) has been added to the Methods section to elaborate on this process. For your convenience, this section has been provided here as well: 

 “There were four primary domains which were examined due to their known associations with suicide risk (Reger et al. 2020): suicide-specific queries, help-seeking queries, general mental health queries, and queries related to financial difficulty. We did not try to be exhaustive by including every possible search term that a user may have utilized within each category, but rather aimed to have 3 to 6 queries representative of each domain. 

Queries were selected for the suicide-specific, general mental health, and financial difficulty categories based upon a review of the literature and use of the “Related Queries” tool available on the Google Trends interface. These finalized categories are as follows: suicide-specific queries (“suicide –squad” [Arora et al. 2016; Barros et al. 2019; Kristoufek et al. 2016; Song et al. 2014], “kill myself,” “kill yourself,” “suicide methods + suicide method,” “commit suicide” [Gunn & Lester 2013], and “fast suicide + easy suicide + quick suicide + painless suicide” [Ma-Kellams et al. 2016]), general mental health queries (“depression” [Arora et al. 2016], “panic attack,” and “anxiety” [Yang et al. 2011; Barros et al. 2019; Arora et al. 2016; Lee et al. 2020]), queries related to financial difficulty (“unemployment” [Arora et al. 2016; Parker et al. 2017], “I lost my job,” “laid off” [Lee 2020], “furlough” (a mandatory temporary leave for employees), and “loan”). The uncategorized query “loneliness” was also included to assess possible impacts of the social distancing interventions implemented to slow the spread of COVID-19.

Queries for the help-seeking category were intended to represent the two most prominent and widely-used crisis services in the United States (National Suicide Prevention Lifeline, the national network of local crisis centers, https://suicidepreventionlifeline.org/; Crisis Text Line, the global organization providing crisis intervention over text message, https://www.crisistextline.org/), as well as the Disaster Distress Helpline, a national hotline providing crisis counseling for those impacted by any natural or human-caused disaster that has been promoted during the COVID-19 pandemic (https://www.samhsa.gov/find-help/disaster-distress-helpline). This finalized category contains the following queries: “crisis text line,” “national suicide prevention lifeline + national suicide hotline,” and “disaster distress helpline + disaster distress hotline.”

Google Trends (https://trends.google.com/trends/?geo=US) allows queries containing certain words to be excluded from the relative search proportion using the “-“ sign, and searches for the popular movie Suicide Squad were thus excluded from our “suicide” search query by requesting data for “suicide –squad” (Ayers et al. 2017). We considered using this feature to filter out searches for “depression” that included “economy,” “economic,” and “are we going into a” to help ensure that we captured searches related to emotional depression rather than an economic depression. However, doing so made no meaningful change in observed search volume so we decided to analyze the simple “depression” query used by Arora et al. (2016), Barros et al. (2019), and Kristoufek et al. (2016). Google trends also allows for the relative search volume associated with several search queries to be combined using the “+” sign. This tool was used to yield a combined relative search proportion for common variations on terms or conceptually linked terms, resulting in the following queries: “national suicide prevention lifeline + national suicide hotline,” “disaster distress helpline + disaster distress hotline,” “suicide methods + suicide method,” and “fast suicide + easy suicide + quick suicide + painless suicide.” “

3. In the results section, it would be useful to discuss the magnitude of your effects (some of which are considerable). At present, this section reports the search terms that had an increase during the period of the study, but not the size of this increase. The variation here is quite large and worthy of comment – it ranges from a 3% increase for “depression” through to over 5700% for “furlough,” yet both of these terms are discussed as if the findings were relatively equal. I would suggest that the magnitude of the effect for some of these terms is a bigger finding than merely the fact that there was an increase. Similarly, the increase in searches for “Disaster Distress Helpline” and “Crisis Text Line” are mentioned in the discussion section as though they are relatively equal, although one saw an increase of 36% and the other and increase of over 3000%.

We present the percentage change in the results. The discussion section has been revised to address these percentage changes in a more nuanced manner and to highlight findings of greater magnitude (see p. 10, l. 245 – p. 11, 256). Specifically, we write:

“Google search behavior appears to have dramatically changed following the emergence of COVID-19 in the United States, particularly for queries related to financial difficulty (“I lost my job,” 226%; “laid off,” 1164%; “unemployment,” 1238%; “furlough,” 5717%) and the Disaster Distress Helpline (“disaster distress helpline + disaster distress hotline,” 3021%). Searches related to suicide are lower than expected, while searches related to general mental health and help-seeking are moderately elevated. Search volumes for suicide-specific terms such as “suicide,” “commit suicide,” and “kill yourself,” which have previously been shown to have a positive correlation with suicide rates (Arora et al. 2016; Gunn & Lester 2013; Ma-Kellams et al. 2016) are all lower than predicted. However, searches for queries representative of financial difficulty have also been shown to have a positive association with suicide rates (Arora et al. 2016; Parker et al. 2017; Lee 2020) and the scale at which search volume for these queries increased in the present study is remarkable.”

The abstract has been revised to reflect these changes as well.

4. It would be useful to explain some of your terms for an international audience – although I understand that your study was focused on America, I was not familiar with some of the terms and services described. In particular, the Disaster Distress Helpline and Crisis Text Line receive some attention in the discussion section, but I was not sure what these services are or how they differ from each other. I also believe that “furlough” may be an American term, and considering its effect size it would be useful to make sure that international readers understand what it means.

We now realize that we have to be more cognizant of the international readership. Furlough has been briefly defined as “a mandatory temporary leave for employees ” (see p. 5, l. 124), and descriptions of the services provided by the Crisis Text Line and the Disaster Distress Helpline have been added to the “Query selection” section of the Methods (see p. 5, l. 127 – l. 134). For your convenience, that description is provided here as well: “Queries for the help-seeking category were intended to represent the two most prominent and widely-used crisis services in the United States (National Suicide Prevention Lifeline, the national network of local crisis centers, https://suicidepreventionlifeline.org/; Crisis Text Line, the global organization providing crisis intervention over text message, https://www.crisistextline.org/), as well as the Disaster Distress Helpline, a national hotline providing crisis counseling for those impacted by any natural or human-caused disaster has been promoted during the COVID-19 pandemic (https://www.samhsa.gov/find-help/disaster-distress-helpline). ”

5. Similarly to my first comment, your discussion section would be greatly strengthened if a little more evidence for your methods was provided. For example, is there evidence to support your claim that a drop in suicide-related terms equates to a drop in suicidality during the study period? This is implied, but if it could be supported by evidence it would make your claims stronger. If not, adding this as a limitation would be worthwhile.

As mentioned in our response to your first comment, several references have been made in the Introduction (see p. 3, l. 87 – p. 4, l. 99) to research which has established associations between Google search behavior and actual suicidal behavior. See our response to your first comment for the summary of the available research included in our Introduction section.

We have also added the following passage to the Discussion section (p. 10, l. 250 – p. 11, l. 260) to emphasize the importance of the changes we observed in queries related to financial difficulty: “Search volumes for suicide-specific terms such as “suicide,” “commit suicide,” and “kill yourself,” which have previously been shown to have a positive correlation with suicide rates (Arora et al. 2016; Gunn & Lester 2013; Ma-Kellams et al. 2016) are all lower than predicted. However, searches for queries representative of financial difficulty have also been shown to have a positive association with suicide rates (Arora et al. 2016; Parker et al. 2017; Lee 2020) and the scale at which search volume for these queries increased in the present study is remarkable. This marked increase in search volume is particularly concerning considering that out of eighty-nine search queries evaluated for correlation with monthly suicide rates in the United States between 2004 and 2017, the strongest positive correlation was found for the query “laid off” with a lag time of 2 months (Lee 2020).” 

Again, we appreciate the thorough review given to our manuscript, and consider the manuscript to be improved because of the reviewers' thoughtful comments.

Thank you for reconsidering the manuscript.

Sincerely,

Madelyn S. Gould, Ph.D., M.P.H.

Irving Philips Professor of Epidemiology in Psychiatry

Columbia University Irving Medical Center

1051 Riverside Drive, Mailbox 72

New York, NY 10032

Tel: (646)774-5763

Fax: (212)543-5281

gouldm@nyspi.columbia.edu

Citations:

Arora VS, Stuckler D, McKee M. Tracking search engine queries for suicide in the United 

Kingdom. Public Health. 2016;137:147-53.

Barros JM, Melia R, Francis K, Bogue J, O’Sullivan M, Young K, et al. The validity of Google 

Trends search volumes for behavioral forecasting of national suicide rates in Ireland. Int. J. Environ. Res. Public Health. 2019 Sept;16(17):3201.

Gunn JF, Lester D. Using Google searches on the internet to monitor suicidal behavior. J. Affect. 

Disord. 2013;148:411-2.

Hagihara A, Miyazaki S. Internet suicide searches and the incidence of suicide in young people 

in Japan. Eur. Arch. Psychiatry Clin. Neurosci. 2012;262:39-46.

Kristoufek L, Moat HS, Preis T. Estimating suicide occurrence statistics using Google Trends. EPJ 

Data Sci. 2016;5(1):32.

Lee JY. Search trends preceding increases in suicide: A cross-correlation study of monthly 

Google search volume and suicide rate using transfer function models. J. Affect. Disord. 

2020;262:155-64.

Ma-Kellams C, Or F, Baek JH, Kawachi I. Rethinking suicide surveillance: Google search data and 

self-reported suicidality differentially estimate completed suicide risk. Clin. Psychol. Sci. 

2016;4(3):480-4.

Parker J, Cuthbertson C, Loveridge S, Skidmore M, Dyar W. Forecasting state-level premature 

deaths from alcohol, drugs, and suicides using Google Trends data. J. Affect. Disord. 2017;213:9-15.

Song TM, Song J, An JY, Hayman LL, Woo JM. Psychological and social factors affecting Internet 

searches on suicide in Korea: a big data analysis of google search trends. Yonsei Med. J. 2014;55(1):254-263.

Yang AC, Tsai S, Huang NE, Peng C. Association of Internet search trends with suicide death in 

Taipei City, Taiwan, 2004-2009. J. Affect. Disord. 2011;132:179-84.

---

## [Decision Letter · Decision Letter 1]

15 Jul 2020

Google searches for suicide and suicide risk factors in the early stages of the COVID-19 pandemic

PONE-D-20-12386R1

Dear Dr. Gould,

We’re pleased to inform you that your manuscript has been judged scientifically suitable for publication and will be formally accepted for publication once it meets all outstanding technical requirements.

Kind regards,

Neal Doran

Academic Editor

PLOS ONE

Additional Editor Comments (optional):

Reviewers' comments:

Reviewer's Responses to Questions

**Comments to the Author**

1. If the authors have adequately addressed your comments raised in a previous round of review and you feel that this manuscript is now acceptable for publication, you may indicate that here to bypass the “Comments to the Author” section, enter your conflict of interest statement in the “Confidential to Editor” section, and submit your "Accept" recommendation.

Reviewer #1: All comments have been addressed

Reviewer #2: All comments have been addressed

2. Is the manuscript technically sound, and do the data support the conclusions?

Reviewer #1: Yes

Reviewer #2: Yes

3. Has the statistical analysis been performed appropriately and rigorously? 

Reviewer #1: Yes

Reviewer #2: Yes

4. Have the authors made all data underlying the findings in their manuscript fully available?

Reviewer #1: Yes

Reviewer #2: Yes

5. Is the manuscript presented in an intelligible fashion and written in standard English?

Reviewer #1: Yes

Reviewer #2: Yes

6. Review Comments to the Author

Reviewer #1: (No Response)

Reviewer #2: (No Response)

7. PLOS authors have the option to publish the peer review history of their article (what does this mean?). If published, this will include your full peer review and any attached files.

Reviewer #1: No

Reviewer #2: No

---

## [Editor Report · Acceptance letter]

17 Jul 2020

PONE-D-20-12386R1 

Google searches for suicide and suicide risk factors in the early stages of the COVID-19 pandemic 

Dear Dr. Gould:

I'm pleased to inform you that your manuscript has been deemed suitable for publication in PLOS ONE. Congratulations! Your manuscript is now with our production department. 

Kind regards, 

on behalf of

Dr. Neal Doran 

Academic Editor

PLOS ONE